# Elevated Interleukin-6 Levels in the Circulation and Peritoneal Fluid of Patients with Ovarian Cancer as a Potential Diagnostic Biomarker: A Systematic Review and Meta-Analysis

**DOI:** 10.3390/jpm11121335

**Published:** 2021-12-09

**Authors:** Hina Amer, Apriliana E. R. Kartikasari, Magdalena Plebanski

**Affiliations:** School of Health and Biomedical Science, Royal Melbourne Institute of Technology, Melbourne, VIC 3000, Australia; S3792283@student.rmit.edu.au (H.A.); magdalena.plebanski@rmit.edu.au (M.P.)

**Keywords:** ovarian cancer, interleukin-6, diagnosis, prognosis, serum, plasma, ascites, tumor microenvironment

## Abstract

Ovarian cancer (OC) is one of the most lethal cancers, largely due to a late diagnosis. This study aimed to provide a comprehensive meta-analysis on the diagnostic performance of IL6 in the blood and ascites separately for advanced and early-stage OC. We included 37 studies with 6948 participants detecting serum or plasma IL6. The plasma/serum IL6 mean level in the late-stage OC was 23.88 pg/mL (95% CI: 13.84–41.23), and the early-stage OC was 16.67 pg/mL (95% CI: 510.06–27.61), significantly higher than the healthy controls at 3.96 pg/mL (95% CI: 2.02–7.73), but not significantly higher than those found in the controls with benign growths in the ovary, which was 9.63 pg/mL (95% CI: 4.16–22.26). To evaluate IL6 in ascites as a diagnostic marker, we included 26 studies with 1590 participants. The mean level of ascitic IL6 in the late-stage OC was 3676.93 pg/mL (95% CI: 1891.7–7146.7), and the early-stage OC was 1519.21 pg/mL (95% CI: 604.6–3817.7), significantly higher than the benign controls at 247.33 pg/mL (95% CI: 96.2–636.0). There was no significant correlation between the levels of circulating and ascitic IL6. When pooling all OC stages for analysis, we found that serum/plasma IL6 provided 76.7% sensitivity (95% CI: 0.71–0.92) and 72% specificity (95% CI: 0.64–0.79). Ascitic IL6 provided higher sensitivity at 84% (95% CI: 0.710–0.919) and specificity at 74% (95% CI: 0.646–0.826). This study highlights the utility of ascitic IL6 for early detection of OC.

## 1. Introduction

Ovarian cancer (OC) is a gynecological malignancies initiated from the ovaries of the female reproductive system [1]. It is one of the most lethal cancers and poses a significant burden to morbidity and mortality in the affected female population [2]. Despite the extensive cytoreductive surgeries and agonizing chemotherapies, survival rates of women with ovarian cancers are still less than 50%. This is because currently there is no effective screening method available to detect OC at its early controllable stage. 

Like most cancers, OC is a heterogenous, multistage disease where an initial localized abnormal tissue growth can spread later to distinct tissues [3]. The triggering agents can be carcinogens present in physical, chemical, or biological forms [4]. A person’s genetic makeup and familial history of OC and breast cancers increase the risk of developing OC [5]. Additionally, unhealthy eating habits, sedentary lifestyles, and aging may also contribute as risk factors [5,6,7,8,9].

The Federation of Gynecology and Obstetrics (FIGO) classified OC to four distinct stages: stage I, stage II, stage III, and stage IV [10]. The staging is a multilevel evaluation of the disease that aids the management plans and predicts the fate of the women with the disease. The early stages, stages I and II, are confined to the ovaries and within the pelvis, respectively. Stages III and IV are late stages involving the spread of cancer cells to various nearby and distinct organs traveling via various physiological channels. The Surveillance, Epidemiology and End Result program (SEER) statistical data in 2018 show that the 5-year survival of women with OC is more than 90% if diagnosed at an early stage compared to less than 20% at a late stage [11,12]. However, the disease is diagnosed at an inevitable late malignant stage in more than 75% of OC cases, mainly due to two reasons. Firstly, patients usually are asymptomatic or present with vague non-specific symptoms at early stages [12]. Secondly, the unavailability of minimally invasive screening procedures, leading patients to undergo invasive surgical biopsies to be correctly diagnosed and stagged [13].

Cancer-specific biomarkers circulating in the blood can be utilized in minimally invasive diagnostic procedures to detect the presence of cancers [13]. In OC, although not specific, Carcinogen Antigen 125 (CA125) and Human Epididymis 4 (HE4) are approved circulating biomarkers used at clinics for OC monitoring and follow-ups [14]. At late stages, CA125 levels directly correlate to tumor burden, and thus it is a reasonable biomarker to follow up patients with post cytoreductive surgery. However, at early stages, CA125 levels may not be increased substantially, possibly due to the low number of tumor cells present, and moreover, its levels can be induced by other physiological conditions including pregnancy, menstruation, and menopause, as well as certain benign conditions such as endometriosis [15]. HE4 is expressed by epithelial cells of the ovaries [16]. HE4 expression is amplified by the increased cancerous growth of ovarian cells, with the excess of HE4 being produced spilling into the circulation. Several studies have proposed HE4 as a superior diagnostic biomarker to CA125, particularly in pre-menopausal women [17]. However, HE4 cannot detect OC at its early stage [18]. Similar to CA125, HE4 expression levels are influenced by pregnancy and aging [19]. According to these observations, the two biomarkers may not be sufficiently accurate for early-stage OC diagnosis [20]. 

Recent studies have highlighted the potential role of cytokines in tumor formation, progression, and metastasis to distant organs [21,22]. The pleiotropic interleukin-6 (IL6) is a cytokine that has been implicated in many cancers [23]. In breast cancer, for example, IL6 is not only involved in its pathogenesis, but it has also been proposed as a diagnostic biomarker and a potential therapeutic target [24]. In OC, IL6 appeared to provide high accuracy to detect stages III and IV [25,26]. Several studies have used IL6 in combination with other circulating biomarkers to provide improved sensitivity (correctly diagnosing a patient with a disease) and specificity (correctly diagnosing a patient without a disease) for OC detection [27]. Additionally, not less than 75% of women with OC developed ascitic fluid in their abdomen. This fluid plays an active role in tumor development and contains diverse acellular fractions including cytokines [28]. IL6 is among the most abundant in ascites, and its concentrations in ascites are a thousand-fold higher than that in the serum [28]. Ascitic IL6 thus has the potential to be used as a minimally-invasive diagnostic biomarker for OC. Biomarkers with high sensitivity and specificity that facilitate early detection will improve quality of life by increasing progression-free and overall survival of OC patients [27,29]. 

Here, we aimed to determine the utility of serum/plasma and ascitic IL6 for the detection of OC separately at early and late disease stages. To that end, we perform a comprehensive meta-analysis on the diagnostic performance of IL6 with an interest to see whether IL6 has the potential to detect not only the late-stage, but also the early-stage OC. 

## 2. Materials and Methods

### 2.1. Study Design 

This study aimed to strategically evaluate information from published studies with meta-analysis, synthesize the pattern of IL6 levels during OC development, and evaluate the evidence of the diagnostic utility of IL6 for OC. The research was carried out by the authors through authentic scientific databases and in agreement with statements provided by “Preferred Reporting Items for Systemic Reviews and Meta-Analysis” (PRISMA) [30].

### 2.2. Search Strategy

The literature was searched systematically in PubMed and Embase databases up to November 2021 for studies investigating the associations between biomarkers IL6 and ovarian cancer. The text word search included (“IL-6”, “IL6”, or “Interleukin-6”), (“ovary”, or “ovarian”), and (“cancer, cancers, carcinoma”, “tumor”, “neoplasm”, “malignant”, or “malignancy”). The selection of articles for studies was based on defined inclusion and exclusion criteria. Titles and abstracts were first manually screened by two authors (H.A. and A.K.). The eligible studies included were peer-reviewed when the full text is available. No restriction to time or age was applied. Additional search by scanning the reference lists from related articles was also performed.

In vitro studies and studies using animal models were set as exclusion criteria. Case studies, previous meta-analyses, reviews, conference papers, and unpublished articles were excluded. Duplicates were removed using endnote software, and articles with full available text and relevant results were included for meta-analysis. Relevant articles were then manually reviewed by two authors (H.A. and A.K.) and selected on the basis of the content of the articles. The inclusion criteria included (1) the study being on ovarian cancer, and (2) the biomarker of interest being soluble IL6 in serum/plasma or peritoneal fluid. 

### 2.3. Quality Assessment

An article’s quality was assessed by noting the author, year, abstract, and number of citations. Moreover, blood handling, ascitic fluid extraction methods, storage, transportation, and use of appropriate statistics for data analysis in method sections of the articles (where available) were observed to assess experimental quality maintained by the researchers. For study fitness, the updated STrengthening the Reporting of OBservational studies in Epidemiology (STROBE) checklist was used to check the quality assessment of the included articles [31].

### 2.4. Data Collection and Extraction

Two reviewers (H.A. and A.K.) recorded data from all studies that met inclusion criteria using a standardized data collection procedure. The author’s name, country, year, and type of study were taken (Appendix A). Total patients and further division of healthy and benign controls were recorded. Patients truly suffering from ovarian cancer were recorded as malignant. To evaluate early and late stages, we organized all the patients under 4 types of FIGO stages. Stage I and II were considered early stages, and III and IV as late stages, as most studies provided the combined values. The IL6 levels in serum/plasma and ascitic fluid were recorded. Similarly, other parameters such as sensitivity and specificity from the receiver operating characteristics (ROC) analysis were recorded. 

### 2.5. Data Analysis and Statistics

The changes in IL6 levels in various OC stages were mapped by determining the pooled mean values of IL6 concentrations in serum/plasma and ascites from the controls and diverse OC stages. The data were effectively separated into 4 groups: healthy, benign, early-stage OC, and late-stage OC. To acquire the pooled mean value, we obtained the mean and standard deviation (SD) of IL6 concentrations from selected studies. When median values were recorded, we estimated the mean values following Cochrane recommendations, with established optimal formulation [32]. We then weighted the mean from each study on the basis of the sample size. As IL6 is expected to display log beta distribution, we constructed the 95% CI following log-transformed data. We applied the Tukey test for multiple comparison analysis. We then back-transformed the variables that had been log-transformed for reporting purposes. Additionally, we performed a linear regression analysis in log space to evaluate the correlation between serum/plasma and ascitic IL6 concentrations. 

To further evaluate the diagnostic utility of IL6, we extracted the sensitivity and specificity values from the selected studies. We back-calculated the true positive (FP), true negative (TN), false positive (FP), and false negative (FN) on the basis of the sample sizes [33]. The formulas used for the calculations are as follows: sensitivity: TP/(TP + FN), specificity: TN/(FP + TN). The overall sensitivity and specificity values were obtained using a random effect model at 95% CI. The between-study heterogeneity was calculated using the I^2^ test. We generated forest plots using OpenMeta (Analyst)^®^V.12.11.14 (Brown University, Providence, RI, USA) [34]. 

## 3. Results

### 3.1. Study Selection and Characterization

The PRISMA flow chart (Figure 1) shows the number of studies searched from PubMed and Embase. A total of 2189 articles were identified. The duplicates of 1084 articles were removed from the selection, and 1105 records were thus selected for screening. After reading the title and the abstract, we removed 286 animal studies, 222 secondary articles, 453 in vitro studies, and 73 conference articles. After removing these articles, 71 studies were included for complete analysis. Then, following our inclusion criteria, the final 55 articles involving IL6 assessed in serum and ascitic fluid of ovarian cancer patients were added for the meta-analysis. With further detailed evaluation, we were able to retrieve 28 studies for IL6 levels only in serum, 15 only in ascites fluid, and 12 both in serum and ascites fluid. All studies are listed in Table 1.

### 3.2. STROBE Checklist

All the articles were assessed for their fitness following the STROBE checklist criteria. It includes author, country, year of publication, introduction describing the background of the study, study design, care taken while handling samples, and valid results through using valid statistical tools. All selected articles passed the STROBE checklist criteria (Appendix A). These include the articles of Sanna et al. [35], Micheli et al. [36], de Lima et al. [37], Rodrigues et al. [38], Kampan et al. [25], Wertel et al. [39], Zhang et al. [40], Crispim et al. [41], Shi et al. [42], Li et al. [43], Dalal et al. [44], Han et al. [45], Kumar et al. [46], Martins-Filho et al. [47], Cantón-Romeo et al. [48], Sanguinette et al. [49], Hao et al. [50], Mikuła-Pietrasik et al. [51], Chudecka-Gɫaz et al. [52], Block et al. [53], Matsuo et al. [54], Ose et al. [55], Lane et al. [56], Trabert et al. [57], Cândido et al. [58], Poole et al. [59], Dobrzycka et al. [60], Matte et al. [61], Autelitano et al. [27], Clendenen et al. [62], Sen et al. [63], Yigit et al. [64], Edgell et al. [65], Nowak et al. [66], Napoletano et al. [67], Tsai-Turton et al. [68], Amonkar et al. [69], Macciò et al. [70], Lutgendorf et al. [71], Kavask et al. [72], Lambeck et al. [73], Chechlinska et al. [74], Costanzo et al. [75], Macciò et al. [76], Gorelik et al. [77], Daraї et al. [78], Dobryszyca et al. [79], Tempfer et al. [80], Scambia et al. [81], Plante et al. [82], Schrӧder et al. [83], van der Zee et al. [84], Moradi et al. [85], Gastl et al. [86], and Berek et al. [87] (Appendix A).

### 3.3. IL6 Levels in the Circulation and Ascites

To evaluate the involvement of IL6 in OC staging and its potential to be used as a diagnostic biomarker, we selected published papers that provide us with the data on IL6 concentrations in serum/plasma and/or ascites from stage-specific OC patients and healthy or benign controls. All selected articles used immuno-based assays including enzyme-linked immunosorbent assay (ELISA) as a method of detection. For plasma/serum IL6 concentrations, we extracted data from a total of 37 studies with 6948 participants. The weighted pooled mean values of IL6 in patients with late-stage OC at 23.88 pg/mL (95% CI: 13.84–41.23) were significantly higher (*p*-value < 0.001) than that of the healthy controls at 3.96 pg/mL (95% CI: 2.02–7.73). Interestingly, the weighted mean values of IL6 in patients with early-stage OC at 16.67 pg/mL (95% CI: 510.06–27.61) were also significantly higher than that of the healthy controls (*p*-value < 0.05). However, when compared to the weighted pooled mean values of the benign controls at 9.63 pg/mL (95% CI: 4.16–22.26), although the trend was similar to the comparison with the healthy controls, the differences were not significant for both the late-stage as well as the early-stage OC (*p*-values of the difference are 0.225 and 0.650, respectively) (Figure 2, Table 1). This result shows the potential of serum/plasma IL6 concentrations as diagnostic biomarkers to detect OC at both early and late stages when compared to the healthy controls. However, the serum/plasma IL6 levels may not be accurate to distinguish between benign and OC growth.

We also gathered 26 studies with 1590 participants in order to evaluate the changes in IL6 levels in the peritoneal fluids from women with OC compared to the benign controls. The weighted pooled mean value of IL6 levels in the ascites from the late-stage OC was 3676.93 pg/mL (95% CI: 1891.7–7146.7), while from the early-stage was 1519.21 pg/mL (95% CI: 604.6–3817.7). Both mean values were significantly higher than the benign controls at 247.33 pg/mL (95% CI: 96.2–636.0) (*p*-value < 0.001 and <0.05, respectively) (Figure 3, Table 2). This result indicates that ascitic IL6 is a useful diagnostic biomarker to detect both the late-stage as well as the early-stage OC.

We further evaluated the correlation between serum/plasma and ascitic IL6 levels in order to determine the potential of using serum/plasma IL6 concentrations as a surrogate marker predicting the levels of IL6 in ascites. We extracted the data from studies that measure IL6 concentrations in both serum/plasma and ascites from the exact same participants. We performed linear regression analysis to evaluate the correlation (Figure 4). We noticed the trend increase in the combined mean values of serum/plasma as well as ascitic IL6 on the basis of OC staging (Figure 4). However, due to the large spread of IL6 values, there was no correlation observed between serum/plasma IL6 levels and ascitic IL6 levels (*p*-value = 0.84, R^2^ = 0.00193), suggesting that the levels of IL6 in the circulation cannot be used as a surrogate marker to predict IL6 levels in peritoneal fluid. Figure 4 also shows that the increased levels of proinflammatory IL6 in peritoneal fluid from the OC groups are 100 times higher than that in the circulation, indicating the involvement of inflammation in OC development. 

### 3.4. Sensitivities and Specificities

Here, we extracted data from 12 selected articles that provide sensitivity and specificity values for IL6 as a diagnostic biomarker for across various OC stages (Appendix A). In serum/plasma, in comparison to the benign controls, the overall values of sensitivity and specificity for IL6 to detect OC of various stages were 76.7% (95% CI: 0.71–0.92) and 72% (95% CI: 0.64–0.79), respectively. In ascites, the overall sensitivity and specificity were 84% (95% CI: 0.710–0.919) and 74% (95% CI: 0.646–0.826), respectively (Table 3, Figure 5). The heterogeneity levels of sensitivity and specificity values described as τ^2^ (Table 3) showed low heterogeneity between data in all chosen studies. These results thus indicate favorable accuracy for both serum/plasma and ascitic IL6 to predict OC at various stages. This meta-analysis result, however, did not provide direct evidence for the utility of IL6 as an early OC detector, as the data came from the combination measures of both early and late OC stages.

Interestingly, Gorelik et al. (2005) shows that IL6 as a single serum biomarker provided 84.1% sensitivity and 86% specificity in predicting OC exclusively at the early stages in comparison to benign controls. Furthermore, in combination with other markers, two studies showed that IL6 provides accurate prediction for the presence of OC at early stages. The first study was by Gorelik et al. (2005), showing that IL6 combined with CA125, GCSF, EGF, and VEGF provided 84.1% sensitivity and 75.7% specificity. The second study was by Han et al. in 2008, showing that when IL6 was combined with CA125, HE4, and ECAD, 84.2% sensitivity and 95.7% specificity were achieved (Table 3). These studies thus suggest that circulating IL6 has an additional value to predict OC at the early stages when combined with other markers.

## 4. Discussion

OC remains to be the most lethal cancer of the reproductive system, with an overall 5 year survival rate of only 46%, due to late diagnosis in more than 75% of the patients. In advanced or late stages, women may respond to initial treatment. However, as the recurrence rate is very high, the late-stage disease leads to a very low survival rate. Thus, early diagnosis of this disease is the key to improving survival from OC. 

As there is currently no effective screening procedure for OC, in addition to its vague symptoms, more research needs to be done to tackle the difficulties in diagnosing this disease. Thus, knowledge in the etiology and mechanism of OC may shed light on both better diagnosis and treatment for this disease. It has been frequently proposed that inflammation, orchestrated by cytokines, may be the driver that induces OC [22]. Within the tumor microenvironment, the OC cells and other stromal cells secrete proinflammatory cytokines, such as IL6 and others. Several studies have further shown that increased inflammation levels in the tumor microenvironment promote poorer prognosis for OC (reviewed in [22,29]). Here, we analyzed published articles in Medline and Embase databases that measured IL6 in the circulation and the peritoneal fluid of women with OC in order to provide fundamental baselines on the absolute concentration levels of IL6 in various stages of this disease, as well as to determine the potential of IL6 as a biomarker for not only the late-stage, but also for the detection of the early-stage OC, which is currently difficult to diagnose. By calculating the pooled weighted mean values followed by post hoc tests, we found that IL6 levels in the circulation were increased in both early- and late-stage OC in comparison to healthy and benign controls. However, the increased plasma/serum IL6 levels were only significant when OC samples were compared to the healthy control samples. This shows that IL6 has the potential to be a diagnostic biomarker to detect OC formation, but it may not be specific enough to distinguish it from benign growth. In contrast, we show a significant increase of IL6 concentrations in the ascites of OC patients, both at the early and late stages, in comparison to benign controls. Thus, this highlights the utility of ascitic IL6 to detect OC importantly at the early stages. 

Using a random effect forest plot, we further determined the overall sensitivity and specificity of IL6 in detecting OC from the benign controls. This part of the analysis was conducted in combined stages of OC. The overall sensitivity and specificity levels of both plasma/serum and ascitic IL6 were favorable to discriminate all-stage OC cases from benign controls. This indicates a strong potential for IL6 to be used as a biomarker for OC. However, with this result, the accuracy of IL6 to detect OC specifically at the early stages is still unknown. Two articles showed high sensitivity and specificity provided by serum/plasma IL6 when combined with other markers to detect early-stage OC; however, more studies are needed to confirm these findings [45,77]. 

In summary, on the basis of our statistical analysis of the combined published data, we found that the levels of circulating and ascitic IL6 are increased with increased stage of OC. The forest plot analyses further showed high accuracy for both plasma/serum and ascitic IL6 to predict OC of mixed stages from the benign controls. For early detection, however, ascitic IL6 from the early-stage OC turned out to be significantly higher than the benign controls, indicating its strong potential as an independent biomarker for early detection of OC. Interestingly, there was no significant correlation between serum/plasma and ascitic IL6, suggesting that IL6 in ascitic fluids may come from other sources and may have distinct functions in the ovarian cancer microenvironment for cancer development. The observation of increased IL6 levels in OC ascites at levels 100 times higher than the serum/plasma levels further suggests IL6 involvement in OC progression within the tumor microenvironment. It has been shown indeed that OC cells secrete IL6 to support its development and metastasis [21]. These observations further suggest that site-specific targeting of the IL6 pathway within the microenvironment may prove to provide additional therapeutic strategies to help control this disease. [87]

## Figures and Tables

**Figure 1 jpm-11-01335-f001:**
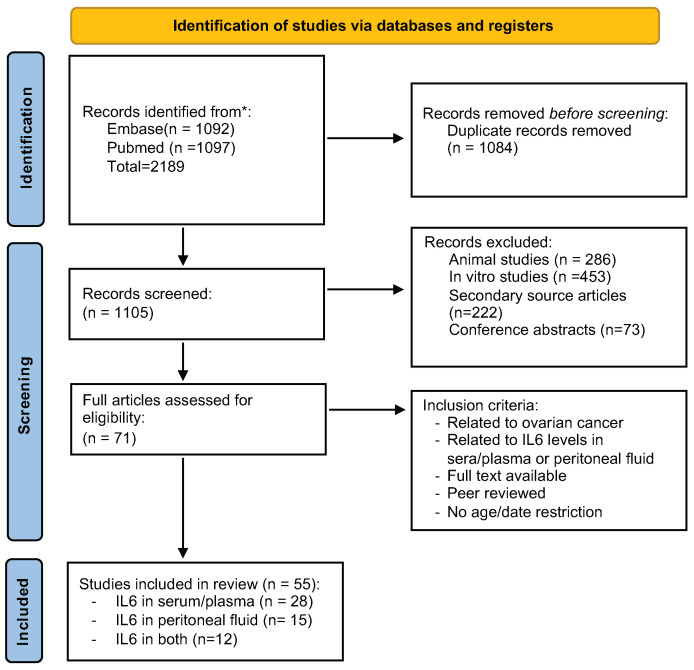
**The PRISMA flowchart of the article selection.** The steps of our study selection are depicted as a flowchart following PRISMA recommendations.

**Figure 2 jpm-11-01335-f002:**
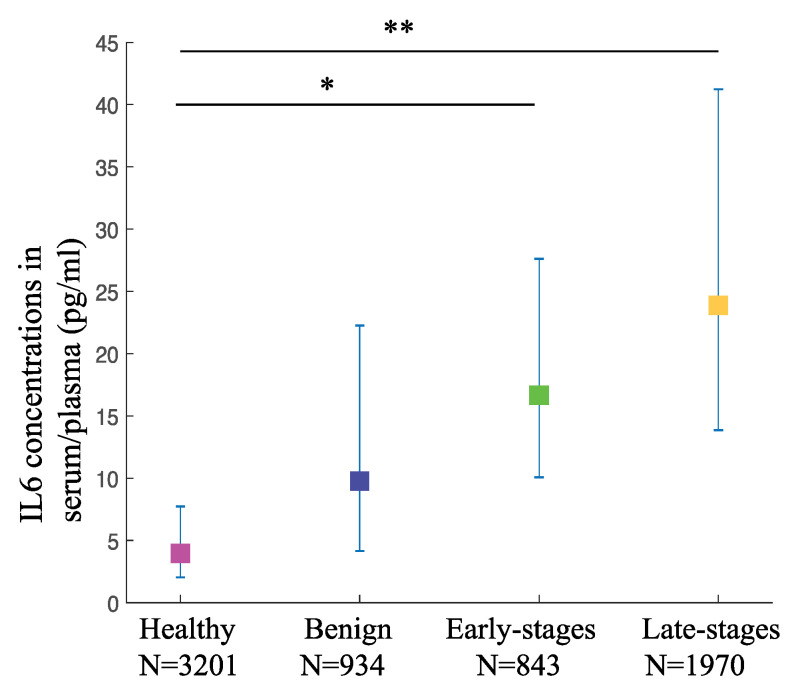
**The circulating IL6 levels are higher in various OC stages compared to the healthy controls.** IL6 levels in serum/plasma from OC and controls are depicted as pooled weighted means ± 95% CI. Multiple comparison analysis was performed by Tukey test. * *p*-value < 0.05, ** *p*-value < 0.01.

**Figure 3 jpm-11-01335-f003:**
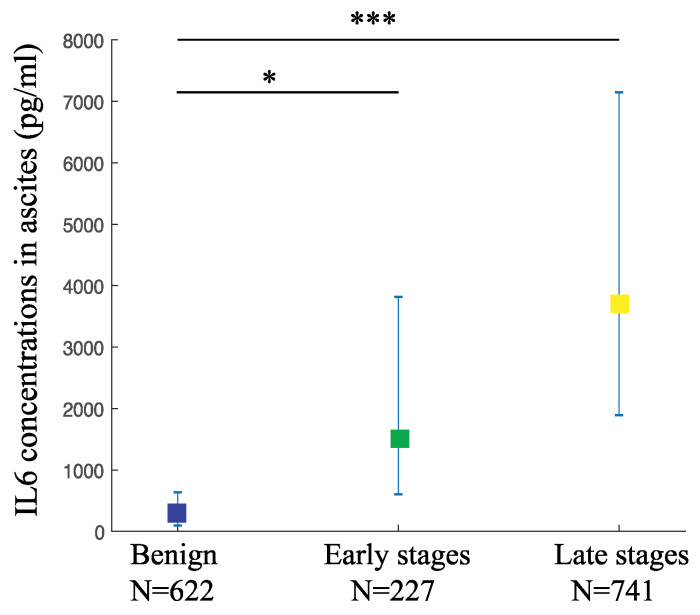
**The ascitic IL6 levels are higher in various OC stages compared to the benign controls.** IL6 levels in ascites from OC and controls are depicted as pooled weighted means ± 95% CI. Multiple comparison analysis was performed by Tukey test. * *p*-value < 0.05, *** *p*-value < 0.001.

**Figure 4 jpm-11-01335-f004:**
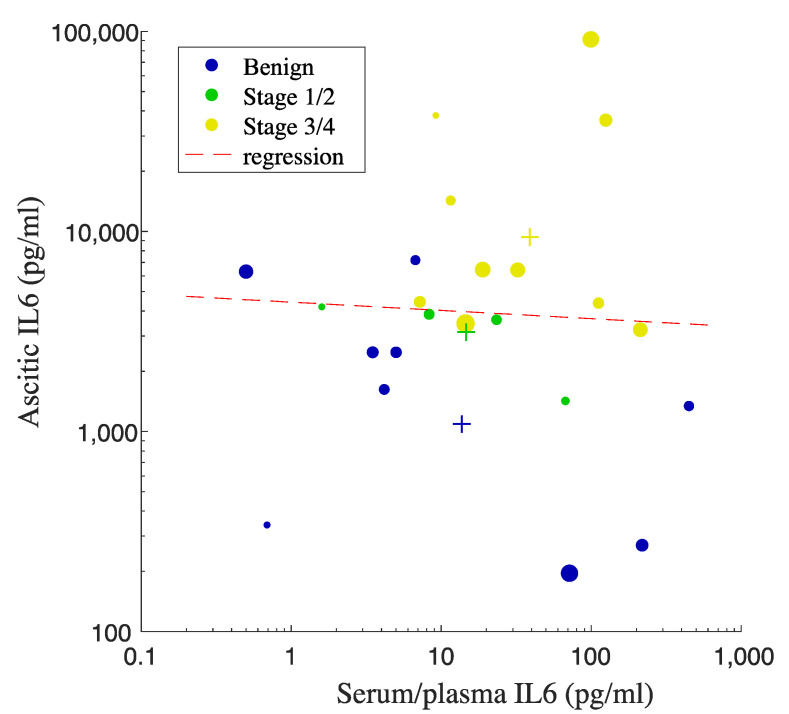
**Linear regression analysis for the correlation between IL6 in serum/plasma vs. ascites**. Each dot represents the mean value, and the size of the dot correlates with the number of participants from each study. Each plus sign indicates the pooled mean value of IL6 from each OC or benign group. IL6 concentrations are in pg/mL The color of the plus signs matches that of the data points.

**Figure 5 jpm-11-01335-f005:**
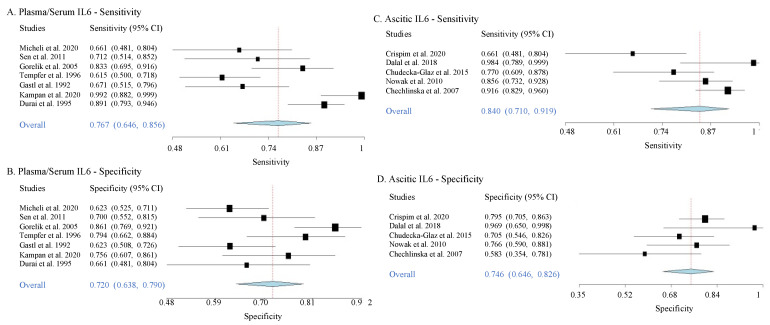
**Forest plots of sensitivity and specificity of plasma/serum and ascitic IL6 to diagnose OC.** The forest plots are generated from the sensitivity and specificity data from various studies when predicting the presence of ovarian cancer (mixed stages) from the benign controls. Each blue diamond indicates the summary of overall sensitivity or specificity from each plot.

**Table 1 jpm-11-01335-t001:** The pooled weighted means of IL6 levels in serum/plasma with 95% CI.

Type	Number of Studies	Pooled Weighted Mean (pg/mL)	95% Confident Interval	*p*-Value (Post-Hoc: OC vs. Healthy Controls	*p*-Value (Post-Hoc OC vs. Benign Controls
Healthy controls	20	3.96	2.02–7.73		
Benign controls	21	9.62	4.16–22.26		
Early-stage OC	20	16.67	10.06–27.61	0.018	0.650
Late-stage OC	20	23.88	13.84–41.23	0.002	0.225

**Table 2 jpm-11-01335-t002:** The pooled weighted means of IL6 levels in ascites with 95% CI.

Type	Number of Studies	Pooled Weighted Mean (pg/mL)	95% Confident Interval	*p*-Value (Post-Hoc OC vs. Benign)
Benign	16	247.33	96.2–636.0	
Early-stage OC	11	1519.21	604.6–3817.7	0.0195
Late-stage OC	13	3676.92	1891.7–7146.7	0.0002

**Table 3 jpm-11-01335-t003:** The overall sensitivity and specificity performance of serum/plasma or ascitic IL6 to correctly indicate the presence of OC from healthy and benign controls.

Type	Number of Studies	Sensitivity				Specificity			
		Overall	95% CI	*p*-Value	τ^2^	Overall	95% CI	*p*-Value	τ^2^
Serum/plasma IL6 for mixed stages	7	76.7%	0.646–0.856	<0.001	0.433	72%	0.638–0.790	<0.001	0.165
Ascitic IL6 for mixedstages	5	84%	0.710–0.919	<0.001	0.492	74%	0.646–0.826	<0.001	0.112
IL6 as one of the combined marker for early stage OC	Gorelik et al., 2005	84.1%	Combined with CA124, GCSF, EGF and VEGF			75.7%	Combined with CA124, GCSF, EGF and VEGF		
	Han et al., 2018	84.2%	Combined with CA125, HE4 and ECAD			95.7%	Combined with CA125, HE4 and ECAD		

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
