# Peer review of "Elevated Interleukin-6 Levels in the Circulation and Peritoneal Fluid of Patients with Ovarian Cancer as a Potential Diagnostic Biomarker: A Systematic Review and Meta-Analysis"

_jpm, 2021, doi:10.3390/jpm11121335_

Round 1

Reviewer 1 Report

manuscript Elevated interleukin-6 levels in the circulation and peritoneal fluid of patients with ovarian cancer as a potential diagnostic biomarker: a systematic review and meta-analysis, Amer et al. meta-analyzed the diagnostic performance of IL-6 based on published studies in Ovarian Cancer.

I think it is a pity that not all studies are inserted and the fact that the study is not in English was an exclusion criteria. Can the author explain if there is a bias due to this?

I would like the definition of valid results from the authors. (line 94)

Maybe some words on the IL-6 detection used in the study would improve the manuscript.

Overall, it is an important study due to the lack of comprehensive biomarker in OC.

Author Response

  • I think it is a pity that not all studies are inserted and the fact that the study is not in English was an exclusion criteria. Can the author explain if there is a bias due to this?

Ans: We have visited all the non-English articles that we have excluded to avoid any bias. Unfortunately, they are all excluded since they are not fulfilling our inclusion criteria. These articles are either in vitro studies, animal studies, not ovarian cancer, not involving measurements of ovarian cancer in the circulation or ascites, or conference abstracts only. We have now updated our manuscript (line 96-98) as well as the result section (line 136-142) and the PRISMA flow chart.

  • I would like the definition of valid results from the authors. (line 94)

Ans: With the word ‘valid’ we meant that the data in the selected papers fitted our inclusion criteria, including that the data are on ovarian cancer, and include IL6 measurements in the circulation and/or ascites. To avoid confusion, we have omitted this word and rewrote the sentence to explain our methodology more accurately (line 96-98).

  • Maybe some words on the IL-6 detection used in the study would improve the manuscript.

Ans: We agree with the reviewer, we have now added a sentence on the results section that describes IL6 detection method used in our selected papers (line 165).

  • Overall, it is an important study due to the lack of comprehensive biomarker in OC.

Ans: We agree with the reviewer, with this paper, we propose that serum/plasma IL6 in combination with other markers can detect early-stage OC. Additionally, IL6 alone in the ascites can be used to detect the presence of OC in women with cysts and developing ascitic fluid.

Reviewer 2 Report

The paper presents a meta-analysis of existing data to determine if levels of IL-6 in the serum/blood and/or ascites could be markers to detect ovarian cancer (OC) early on.

The manuscript requires thorough editing if not re-writing. At points it's difficult to understand the intention of the authors. E.g. line 39: "This confers due to lack of OC screening tools at an early controllable stage in high-risk women," or lines 42-43: "...a person’s genetic makeup is one the most defiant behaviour of malignant cells pathogenesis, management, and prognosis."

In its present form, it's hard to tease out conclusions. Are levels of IL-6 in the serum/blood and/or ascites predictors of early ovarian cancer? The authors say that there is no correlation between serum/plasma levels of IL-6 and ascites' levels. However, both levels correlate with OC staging  (at least for later stages). Am I understanding this correctly? The authors seem to contradict themselves on this point. For example, on line 210: "These results show the favorable accuracy of IL6 in both serum/plasma and ascites to predict all stage OC. These data however did not provide a strong evidence of the utility of IL6 as a diagnostic biomarker for OC 211 at early stages." So did they or didn't they? Is IL-6 better than HE4, mentioned in the introduction?

I think there is a language barrier that might prevent the authors from getting their message across. I would recommend hiring a professional scientific writer or editor and to organize the paper in a clearer fashion. There is data out there already (some of it cited in the paper and used in the meta-analysis) that shows the correlation between IL-6 and stages of OC. This is not new. Did the meta-analysis support these finding? I understand that it did not for the early stages of OC, which is when it would be meaningful to have a diagnostic marker. If this is the conclusion, then this study did not support its hypothesis, which should be stated clearly.

Also, there is no mention of the literature describing plasma levels of IL-6 as a marker of breast cancer, which probably should be included.

Author Response

  • The paper presents a meta-analysis of existing data to determine if levels of IL-6 in the serum/blood and/or ascites could be markers to detect ovarian cancer (OC) early on.The manuscript requires thorough editing if not re-writing. At points it's difficult to understand the intention of the authors. E.g. line 39: "This confers due to lack of OC screening tools at an early controllable stage in high-risk women,"

Ans: The intention is to specify that there are no approved methods for OC screening, thus the sentence has now been rephrased (line 40).

  • or lines 42-43: "...a person’s genetic makeup is one the most defiant behaviour of malignant cells pathogenesis, management, and prognosis."

Ans: This sentence has been rephrased (line 43-44).

  • In its present form, it's hard to tease out conclusions. Are levels of IL-6 in the serum/blood and/or ascites predictors of early ovarian cancer?

Ans: Based on our analyses, the levels of IL6 in ascites but not IL6 in serum/plasma are predictors for early OC. When combined with other markers including CA125 and HE4, IL6 improves these markers detections for early OC. We have added/rephrased this conclusion throughout the manuscript (line 24-26, line 172-174, line 189-190, line 200-208, line 222-224, line 229-230, line 279-304)

  • The authors say that there is no correlation between serum/plasma levels of IL-6 and ascites' levels. However, both levels correlate with OC staging (at least for later stages). Am I understanding this correctly?

Ans: Yes, it is indeed what we found. Due to this comment, we reanalyzed the correlation data and weighted the means based on the number of participants. However, the result was still the same, no correlation was observed. We have now plotted the data as a scatter plot fitted in log space that also shows the mean values from each benign, early, and late-stage group (Figure 4). Although the trend of increase in the mean values of OC stages in this scatter plot (Figure 4) is consistent with the observations of increased IL6 values with OC staging (Figure 2 and 3), due to the large spread of IL6 values, there is no correlation between IL6 levels in plasma/serum and ascites observed, P-value=0.84, R2=0.00193. This result thus suggests that the actual measures of IL6 levels in the circulation are not directly proportional to its levels in the ascites. Thus, it is necessary to measure the two independently. We have rewritten this section (line 200-208, line 300-305)

  • The authors seem to contradict themselves on this point. For example, on line 210: "These results show the favorable accuracy of IL6 in both serum/plasma and ascites to predict all stage OC. These data however did not provide a strong evidence of the utility of IL6 as a diagnostic biomarker for OC 211 at early stages." So did they or didn't they?

Ans: This sentence refers to the forest plot-accuracy study of IL6 when being used to predict the presence of OC that includes mixed early and late stages of the disease, as listed in Suppl. 3. We have now rewritten the sentence, and discuss it also in the discussion section (line 222-224, line 293-295)

  • Is IL-6 better than HE4, mentioned in the introduction?

Ans: As we did not do a side-by-side comparison between IL6 and HE4, we could not answer this question. What we know, based on the previous studies, the combination of IL6 with HE4 and other circulating proteins provides an outstanding combined marker for early detection of OC. We have now rephrased our introduction and discussion parts of our manuscript to reflect this view (line 60-65, line 293-295).

  • I think there is a language barrier that might prevent the authors from getting their message across. I would recommend hiring a professional scientific writer or editor and to organize the paper in a clearer fashion.

Ans: The article has been revised by all authors to get the message across and has now gone through a thorough English revision as recommended by the journal.

  • There is data out there already (some of it cited in the paper and used in the meta-analysis) that shows the correlation between IL-6 and stages of OC. This is not new. Did the meta-analysis support these finding? I understand that it did not for the early stages of OC, which is when it would be meaningful to have a diagnostic marker. If this is the conclusion, then this study did not support its hypothesis, which should be stated clearly.

Ans: Our meta-analysis supports the published data showing the correlation between IL6 and the mixed stage OC. However, what is unique about our findings is indeed the potential of ascitic IL6 as a biomarker for the early stages of OC. The serum/plasma IL6 shows its potential for early-stage OC detection, however, this is only when it is combined with other markers including CA125 and HE4. We have now stated the abstract, result and discussion sections (line 24-26, line 172-174, line 189-190, line 200-208, line 222-224, line 229-230, line 279-304)

  • Also, there is no mention of the literature describing plasma levels of IL-6 as a marker of breast cancer, which probably should be included.

Ans: IL6 has the potential to be a diagnostic biomarker in other cancers as well, including breast cancer. However, since our study was focused to define the significance of IL6 concerning various OC stages, other cancers were not mentioned. However, we agree with the reviewer, to give readers a context to our findings, we add sentences in the introduction section regarding this aspect of IL6 (line 66-68).

Round 2

Reviewer 2 Report

Much improved! Now a very interesting paper.